# Structural inhibition of dynamin-mediated membrane fission by endophilin

**Annika Hohendahl[1†], Nathaniel Talledge[2,3,4,5†], Valentina Galli[1], Peter S Shen[4], Frédéric Humbert[1], Pietro De Camilli[6,7,8,9,10], Adam Frost[2,3,4,5*], Aurélien Roux[1,11*]**

[1]Biochemistry Department, University of Geneva, Geneva, Switzerland; [2]Department of Biochemistry and Biophysics, University of California, San Francisco, United States; [3]California Institute for Quantitative Biomedical Research, University of California, San Francisco, United States; [4]Department of Biochemistry, University of Utah, Salt Lake City, United States; [5]Chan Zuckerberg Biohub, San Francisco, United States; [6]Department of Neuroscience, Yale University School of Medicine, New Haven, United States; [7]Program in Cellular Neuroscience, Neurodegeneration and Repair, Yale University School of Medicine, New Haven, United States; [8]Kavli Institute for Neuroscience, Yale University School of Medicine, New Haven, United States; [9]Department of Cell Biology, Yale University School of Medicine, New Haven, United States; [10]Howard Hughes Medical Institute, Yale University School of Medicine, New Haven, United States; [11]Swiss National Centre for Competence in Research Programme Chemical Biology, Geneva, Switzerland

**\*For correspondence:**
adam.frost@ucsf.edu (AF);
aurelien.roux@unige.ch (AéR)

[†]These authors contributed equally to this work

**Competing interests:** The authors declare that no competing interests exist.

**Abstract** Dynamin, which mediates membrane fission during endocytosis, binds endophilin and other members of the *Bin-Amphiphysin-Rvs* (BAR) protein family. How endophilin influences endocytic membrane fission is still unclear. Here, we show that dynamin-mediated membrane fission is potently inhibited in vitro when an excess of endophilin co-assembles with dynamin around membrane tubules. We further show by electron microscopy that endophilin intercalates between turns of the dynamin helix and impairs fission by preventing *trans* interactions between dynamin rungs that are thought to play critical roles in membrane constriction. In living cells, overexpression of endophilin delayed both fission and transferrin uptake. Together, our observations suggest that while endophilin helps shape endocytic tubules and recruit dynamin to endocytic sites, it can also block membrane fission when present in excess by inhibiting inter-dynamin interactions. The sequence of recruitment and the relative stoichiometry of the two proteins may be critical to regulated endocytic fission.

DOI: https://doi.org/10.7554/eLife.26856.001

## Introduction

Dynamin is a 97 kDa GTPase that mediates membrane fission in several endocytic routes. Dynamin forms tetramers in solution via the stalk domain (*Muhlberg et al., 1997*; *Reubold et al., 2015*) and polymers around membrane tubes (*Sweitzer and Hinshaw, 1998*; *Takei et al., 1995*). Polymeric forms have a GTPase activity increased up to a 1000-fold over basal hydrolysis rates (*Stowell et al., 1999*). This increase is proposed to result from *trans* interactions between GTPase (G) domains of adjacent turns of a dynamin helical polymer (*Chappie et al., 2010*). Constriction of the dynamin polymer is required for membrane fission (*Antonny et al., 2016*), but how GTPase activity drives constriction is still debated. The disassembly model proposes that dynamin assembles into a highly constricted helix in the GTP-loaded state and that GTP-hydrolysis-dependent disassembly is required for fission (*Shnyrova et al., 2013*). In the constriction/ratchet model, dynamin G domains

are proposed to act as motor heads that use GTP hydrolysis energy to drive relative displacement of helical turns and thereby decrease the diameter of the dynamin helix to constrict the underlying membrane tube. Spontaneous fission then takes place at the edge of the constricted tube due to thermal fluctuations (*Morlot et al., 2012*).

In the cell, dynamin is recruited to endocytic vesicle necks just before fission (*Taylor et al., 2011*). The dynamin-binding N-BAR domain containing proteins are proposed to participate in this recruitment (*Ferguson and De Camilli, 2012*; *Taylor et al., 2011*). N-BAR domains are a subfamily of the crescent-shaped *B*IN-*A*mphiphysin-*R*vs (BAR) domains (*Frost et al., 2009*; *Peter et al., 2004*) with an additional N-terminal amphipathic helix. N-BAR proteins sense and induce membrane curvature and are thus proposed to either help induce or bind avidly to the highly curved necks of endocytic pits (*Farsad et al., 2001*; *Peter et al., 2004*; *Takei et al., 1999*). The SH3 domain of N-BAR proteins binds to dynamin's proline-rich domain (PRD) (*Grabs et al., 1997*; *Ringstad et al., 1997*) recruiting dynamin to vesicle necks (*Sundborger et al., 2011*).

N-BAR proteins have been proposed to directly participate in membrane fission, as their N-terminal alpha-helix induces membrane curvature, thus promoting constriction and possibly fission (*Boucrot et al., 2012*; *Farsad et al., 2001*). However, the role of N-BAR proteins in membrane fission is still unclear: the BAR domain is also proposed to act as a stable scaffold, blocking further constriction below its preferred curvature and thereby stabilizing a tubule of a given diameter and inhibiting fission (*Boucrot et al., 2012*). This scaffold was also proposed to hinder lipid diffusion, promoting fission of tubules under pulling forces (*Renard et al., 2015*; *Simunovic et al., 2017*).

The contribution of N-BAR proteins to membrane fission mediated by dynamin is even less clear: amphiphysin was proposed to enhance dynamin-mediated membrane fission (*Takei et al., 1999*), a stimulatory effect later found to be curvature-dependent (*Yoshida et al., 2004*). Early findings showed that endophilin inhibited dynamin-mediated membrane fission (*Farsad et al., 2001*). However, different results were obtained in other assays (*Meinecke et al., 2013*; *Neumann and Schmid, 2013*). Moreover, a fast endocytic pathway strictly depends on endophilinA2, which may indicate that endophilin proteins can facilitate dynamin-mediated fission in certain contexts (*Boucrot et al., 2015*). Because of the differing published results, we sought to measure the direct impact of endophilin on dynamin's oligomeric properties and membrane fission activity.

## Results

### Endophilin inhibits dynamin-mediated membrane fission

To study the effect of endophilin on dynamin-mediated fission, we generated lipid tubules from hydrated membrane sheets and recorded videos of their dynamics after protein addition (*Itoh et al., 2005*; *Roux et al., 2006*). Once formed, we incubated membrane sheets either with dynamin alone (*Figure 1A*) or sequentially with first endophilin and then dynamin (*Figure 1B*). In the latter, fluorescently labeled proteins were used and both were detected on the tubules, confirming their co-assembly (*Figure 1—figure supplement 1A*). The contrast of the endophilin-dynamin tubules was stronger, and these tubules appeared thicker under Differential Interference Contrast (DIC) microscopy, probably reflecting the co-assembly of endophilin and dynamin. Upon GTP addition (t = 0 s), the tubules coated by dynamin alone underwent fission within a few seconds as previously described (*Roux et al., 2006*), while tubules coated by endophilin and dynamin did not break nor alter their shape (see *Figure 1A,B*). We concluded from these experiments that endophilin could inhibit dynamin-mediated membrane fission at the concentrations employed.

To further investigate this inhibitory function, we tested dynamin activity in the presence of endophilin in other functional assays. We first performed a GTPase malachite green assay and found that a fourfold excess of endophilin reduced the GTPase activity of liposome-bound dynamin by half (*Figure 1—figure supplement 1B*), as previously published (*Farsad et al., 2001*). We tested if dynamin still underwent GTP hydrolysis-driven mechano-chemical constriction in the presence of endophilin by visualizing the rotation of beads attached to dynamin-coated lipid tubules after addition of GTP into the chamber (*Roux et al., 2006*) (*Figure 1C*). As previously shown, beads rotated with an average speed of 8–9 rotations/second with dynamin alone, whereas no rotation was observed in the presence of endophilin (*Figure 1D,E*). These results showed that endophilin blocked mechano-

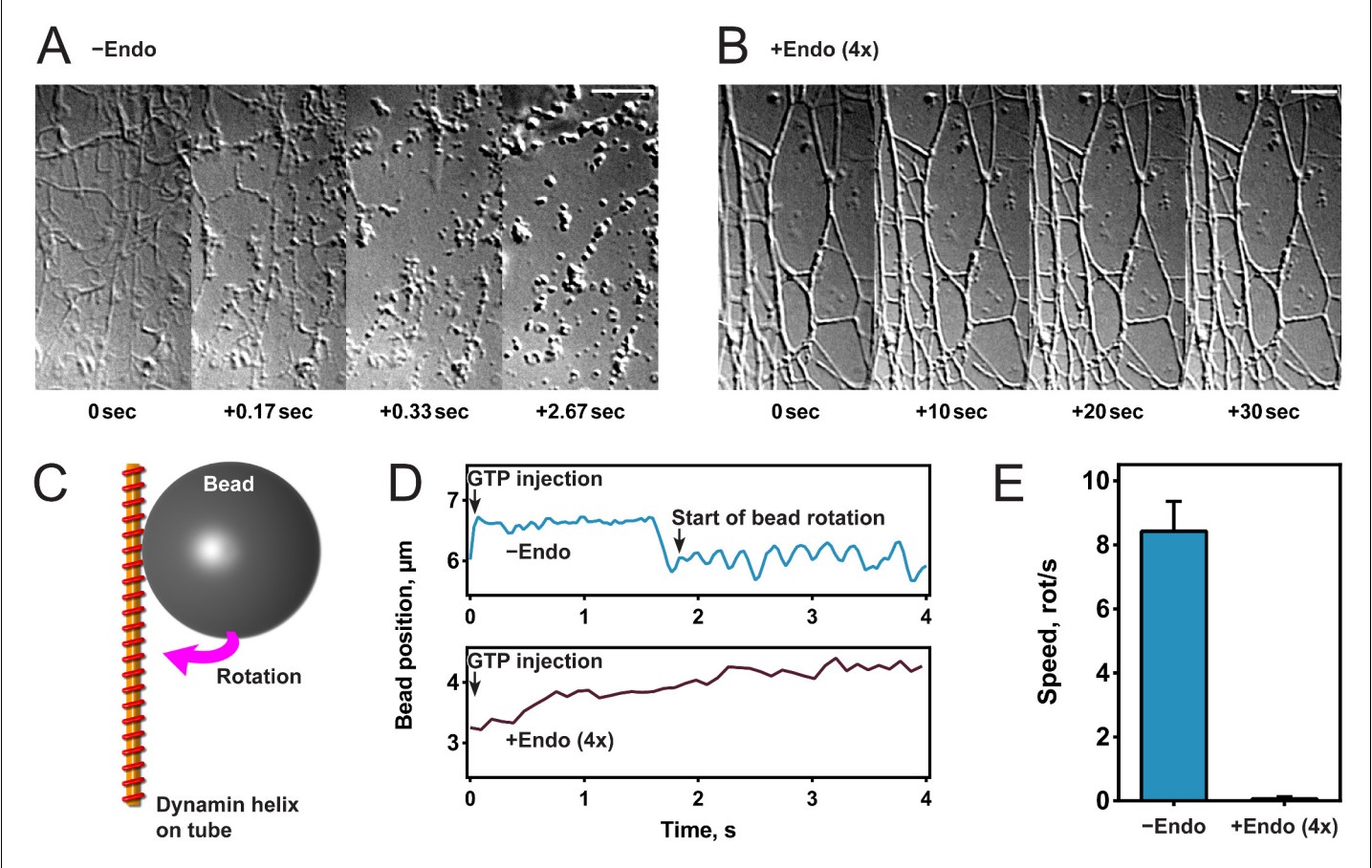

**Figure 1.** Endophilin inhibits dynamin-mediated constriction and fission. (**A, B**) Visualization of tubule fission using membrane sheets assay. We injected GTP at t = 0 s on tubules generated by dynamin (**A**) or dynamin and endophilin (4x) (**B**). (**C**) Scheme of bead rotation assay. (**D, E**) Endophilin inhibits dynamin constriction, as the bead does not rotate in the presence of endophilin. Representative traces of the bead position relative to the tubule axis. Oscillations are caused by the bead rotating around the tubule (**D**), averaged maximal speeds (**E**). Error bars indicate standard deviation. Scale bars, 5 μm.

DOI: https://doi.org/10.7554/eLife.26856.002

The following figure supplement is available for figure 1:

**Figure supplement 1.** Colocalization of dynamin and endophilin and GTPase assay.

DOI: https://doi.org/10.7554/eLife.26856.003

chemical constriction of dynamin and subsequent membrane fission. However, we cannot exclude that dynamin may constrict through another mechanism.

## Endophilin reduces rate and efficiency of membrane fission in a concentration-dependent manner

To better characterize and define the required molecular interactions for this apparent fission inhibition, we used an assay that enabled us to measure the rate of dynamin-mediated membrane fission (*Figure 2A*). We held giant unilamellar vesicles (GUVs) under aspiration into a glass micropipette and pulled lipid nanotubes out from the GUV using optical tweezers (Materials and methods and *Morlot et al., 2012*). We injected dynamin, endophilin, and GTP simultaneously a few microns away from the membrane tube via separate injection pipettes allowing for various sequences of injection during time-lapse confocal imaging. The fission time – which is the inverse of the fission rate – was defined as the time between the start of the injection and the breakage of the tube (*Figure 2C*). We measured fission times and plotted cumulative probability curves for various endophilin/dynamin molar concentration ratios, with the concentration of dynamin held constant at 5 μM. Exponential

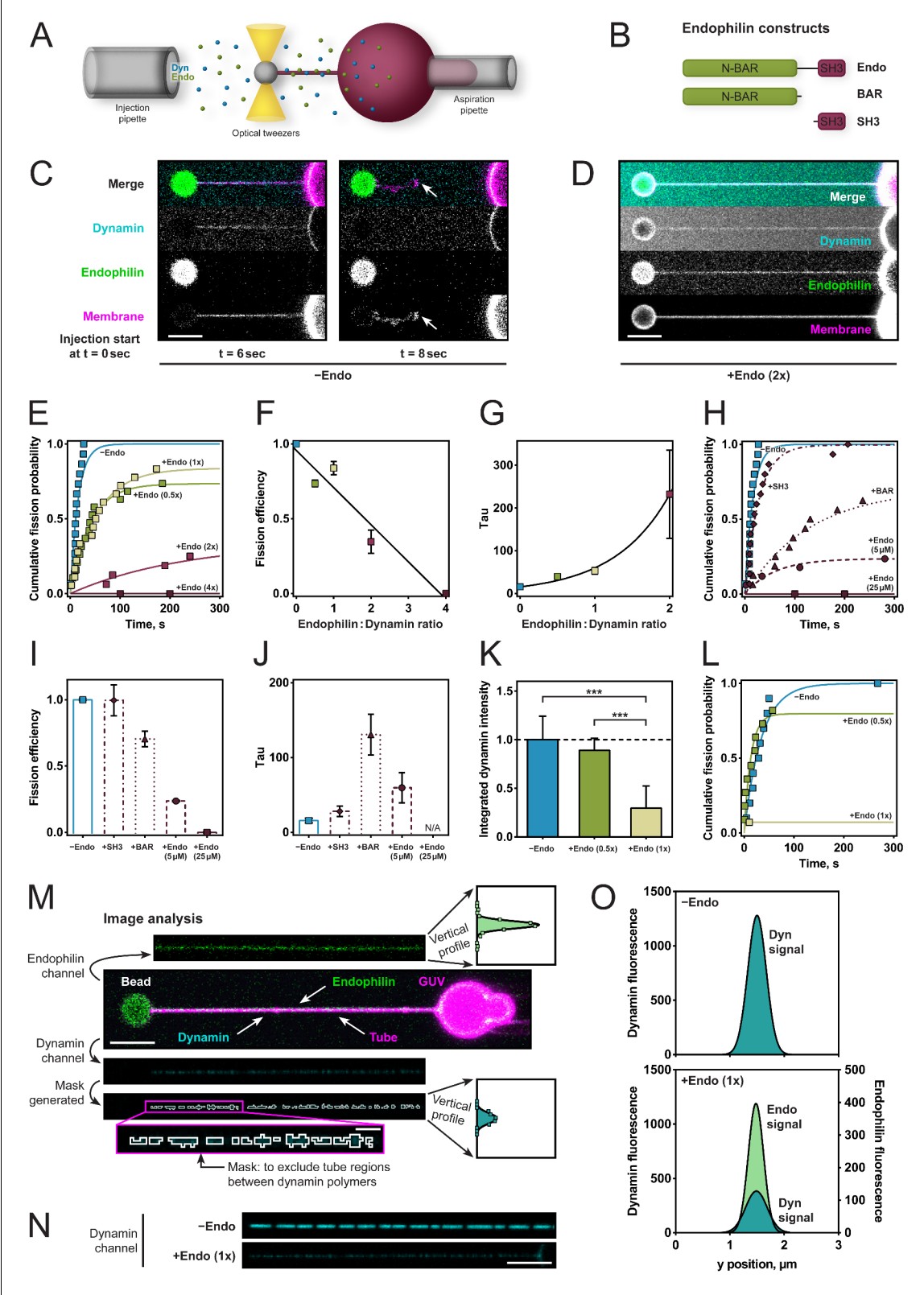

**Figure 2.** Endophilin reduces fission rate, fission efficiency and dynamin density. (**A**) Tube pulling setup. A membrane tube is pulled from a GUV aspirated in a micropipette. Protein is injected using a second pipette. (**B**) Endophilin constructs used for the experiments. (**C**) Confocal images of tube after dynamin injection, as in setup (**A**). GUV is on the right, bead is on the left. (**D**) Confocal images of a tube after co-injection of endophilin (2x) dynamin (1x) and GTP. (**E**) Cumulative fission probability of tubes for various molar ratios of endophilin/dynamin, using 5 µM dynamin, 150 µM GTP.

*Figure 2 continued on next page*

*Figure 2 continued*

Lines: exponential fits to a*(1-exp(-t/τ)), n(−Endo)=15, n(+Endo (0.5x))=19, n(+Endo (1x))=18, n(+Endo (2x))=16, n(+Endo (4x))=15. Matlab code available in Source code file 1. (F) Fission efficiencies extracted from fits to data shown in (E). (G) Average fission time τ from fits to data shown in (E). (H) Cumulative fission probability for different endophilin constructs shown in (B) at 4x molar endophilin/dynamin ratio. n(+Endo (5 μM))=17, n(BAR)=16, n (SH3)=15. Lines: exponential fits to a*(1-exp(-t/ τ), Matlab code available in Source code file 1. (I) Fission efficiency extracted from fits to data shown in (H). (J) Average fission times τ extracted from fits to data shown in (H). Error bars in (F, G, I, J) indicate 95% confidence intervals of fits. (K–O) Decreased dynamin fluorescence density in co-complex with endophilin correlates with reduced fission efficiency. (K) Averaged dynamin fluorescence intensities for three different endophilin/dynamin ratios. The indicated values are calculated from integrals of the fluorescence peaks obtained following the image analysis explained in M. Source data are available in the *Figure 2—source data 1*. (L) Cumulative fission probability for the same tubes whose fluorescence was measured in (K). Matlab code available in Source code file 2. For (K–O), 150 μM GTP and 5 μM Dyn were used. (M) Image analysis of tubes coated with either dynamin alone or in co-complex with endophilin. From the dynamin image, a mask is generated (see magenta box), and a projection along the edge perpendicular to the tube axis is calculated (see vertical profiles). The intensity values shown in (K) are integrals of these vertical profiles. (N) Representative confocal images of dynamin fluorescence intensity signal for dynamin alone (−Endo) or in co-complex with endophilin (+Endo (1x)). (O) Vertical profiles obtained by image analysis shown in (M) of tubes shown in (N). Error bars in (K) indicate standard deviation. Scale bars are 5 μm, except in magenta box for (M), 1 μm.

DOI: https://doi.org/10.7554/eLife.26856.004

The following source data and figure supplements are available for figure 2:

**Source data 1.** Dynamin fluorescence intensities for three different endophilin/dynamin ratios.

DOI: https://doi.org/10.7554/eLife.26856.006

**Figure supplement 1.** Endophilin intensity on tube increases with increased endophilin concentration in solution.

DOI: https://doi.org/10.7554/eLife.26856.005

**Figure supplement 1—source data 1** Quantification of endophilin signal intensity on the same tubes as those whose dynamin signal was measured for *Figure 2K*.

DOI: https://doi.org/10.7554/eLife.26856.007

fits to a*(1-exp(-t/τ)) (raw data and fitting available upon request) revealed the fission efficiency (*a*) and characteristic time (τ) for each condition, incl. 95% confidence intervals for the two fit parameters. Fission efficiency, *a*, corresponds to the fraction of uncut tubes. At low endophilin/dynamin ratios (0.5x/1x), fission was delayed (39.5 ± 3.8 s for ratio 0.5x and 52.8 ± 7.8 s for ratio 1x, compared to 15.8 ± 3.7 s for dynamin alone; *Figure 2E,F,G*). At higher endophilin ratios, where both dynamin and endophilin are bound to the tube (*Figure 2D*), membrane fission was dramatically inhibited (2x) or completely inhibited (4x) (*Figure 2E,F,G*).

To test whether direct interactions between dynamin and endophilin and between endophilin and the membrane were required for this inhibition, we examined the effect of endophilin's N-BAR and SH3 domains alone on dynamin-mediated fission rates and efficiencies at the highest (4x) endophilin-dynamin concentration ratio (*Figure 2B*). No difference in fission efficiency of dynamin was observed for the SH3 domain (*Figure 2H,I*). By contrast, the N-BAR domain alone significantly decreased the fission rate (*Figure 2H,J*), while fission efficiency was reduced only slightly (*Figure 2H,I*).

The inhibitory effect was also present when the reactions had the total protein concentration kept constant across the various protein ratios. The fission inhibition was slightly less pronounced than when the dynamin concentration was kept constant at 5 μM, but the overall inhibitory effect remained (*Figure 2H,I,J*). This was a critical observation as changing the concentrations of dynamin and endophilin affects their ability to bind on these membrane tubules (*Roux et al., 2010*; *Sorre et al., 2012*).

## Co-assembly with endophilin correlates with fission inhibition

We next wondered how endophilin inhibits dynamin's fission activity. The distance between turns of dynamin is increased by a factor of two in the presence of endophilin, as seen by electron microscopy (EM) (*Farsad et al., 2001*). We hypothesized that this morphological difference of the dynamin coat induced by endophilin might be due to the intercalation of endophilin between the dynamin rings. Such a molecular intercalation or interleaving could, for example, block the molecular interactions that form in trans between G domains of two adjacent helical turns. These *trans* G domain interactions have been proposed to be responsible for GTPase-dependent conformational changes and constriction, thus precluding G domain interactings may inhibit fission (*Chappie et al., 2010*; *Chappie et al., 2011*).

To test the hypothesis that the co-assembled architecture of the coat was responsible for fission inhibition, we assumed that reduced dynamin density would reflect the incorporation of endophilin within the protein coat, so we looked for a way to monitor fission rate as a function of dynamin density on the lipid tubule. We, therefore, used a lipid tube pulling assay, and tubes were decorated by co-injection of dynamin and endophilin (*Figure 2M*). We assessed formation of the co-complex by measuring the fluorescence signals of both proteins and then injected GTP and measured the time to fission. The dynamin signal was lower when endophilin was co-injected, as compared to dynamin alone (*Figure 2K,N*), suggesting that endophilin was intercalating within the dynamin coat. The endophilin signal on the tube increased, as expected, with increasing endophilin concentration (*Figure 2—figure supplement 1*). The dynamin fluorescence reduced by 2–3 fold (*Figure 2K,O*). Fission activity was dramatically reduced on tubes with the endophilin-dynamin co-complex, whereas it proceeded normally on tubes coated with dynamin only (*Figure 2L*). Our results thus support the idea that the molecular arrangement of the endophilin-dynamin co-complex was responsible for fission inhibition.

## Endophilin binds to the membrane in-between dynamin rungs

To understand the molecular nature of the inhibition, we sought to study the structure of the endophilin-dynamin complex by EM. However, currently available EM images do not have sufficient resolution to distinguish BAR domain proteins and turns of the dynamin helix (*Farsad et al., 2001*; *Shupliakov et al., 1997*; *Takei et al., 1998*). We thus prepared lipid tubules coated by wild type and full-length endophilin, dynamin, and endophilin plus dynamin for negative stain and cryoEM imaging, hoping to resolve the relevant molecular arrangements (*Figure 3*). In contrast with the endophilin-only and dynamin-only oligomers (*Figure 3A–B*), we observed that the endophilin-dynamin co-complex suffered from highly variable spacing between turns of what appeared to be discrete dynamin oligomers (29.4 ± 5.2 nm, *Figure 3C*). Since this variability prevented high-resolution 3D reconstructions, we focused instead on resolving the nearest-neighbor arrangements within the co-complex by 2D averaging.

We extracted apparent asymmetric units of the membrane-bound proteins projected perpendicular to the long axis of the membrane tubule for reference-free 2D classification (*Figure 3A–C*, inset circles). Consistent with prior structural work, this approach resolved a uniform coating of endophilin-only with a ~5 nm spacing between turns (*Figure 3A*, [*Mim et al., 2012*]). Dynamin bound to GDP or GMPPCP also formed well-ordered helices on membranes. Averaging resolved the 'T-shaped' dynamin dimer, *trans* interactions between adjacent G-domains in the GMPPCP-bound state, and the phosphoinositide-binding pleckstrin homology (PH) domain (*Chappie et al., 2011*; *Zhang and Hinshaw, 2001*) (*Figure 3B* and *Figure 3—figure supplement 1*). Further analysis of the dynamin-only GMPPCP tubules revealed a mixture of 1-start (34%) and 2-start (66%) helices (*Figure 3—figure supplement 1*). The stalk-to-stalk spacing between wild-type, full-length dynamin turns was nearly identical for the 1-start (13.9 nm) versus 2-start (14.0 nm) helices and is consistent with prior structural work using the dynamin truncations that lack the C-terminal proline-rich domain (*Figure 3—figure supplement 1*) (*Antonny et al., 2016*; *Chappie et al., 2011*).

As previously reported (*Farsad et al., 2001*; *Itoh et al., 2005*; *Sundborger et al., 2011*; *Takei et al., 1998*; *Takei et al., 1999*), turns of a dynamin oligomer were spaced ~2 x further apart in the endophilin co-complex in either or GDP or GMPPCP when compared with dynamin alone when visualized with cryoEM or negative stain EM (*Figure 3B–G*). Consistently, cryoEM and 2D averaging revealed multiple endophilin molecules apparently interleaved between adjacent dynamin oligomers (*Figure 3C*), effectively separating the G domains. In this arrangement, when both endophilin and dynamin are bound to the membrane and to each other via SH3-PRD domain interactions, dynamin molecules from adjacent turns are too far apart to interact in trans.

## Endophilin overexpression blocks endocytic pits in vivo

One prediction of these in vitro observations is that an excess of endophilin at coated pits should delay the fission of dynamin-dependent endocytic pits. We tested this by overexpressing endophilin in cells stably expressing dynamin-GFP and analyzing them through Total-Internal-Reflection-Fluorescence-Microscopy (TIRFM). SKMEL genome-edited cells (a gift of David Drubin) express dynamin-GFP at endogenous levels, which allowed us to manipulate the endophilin/dynamin ratio by

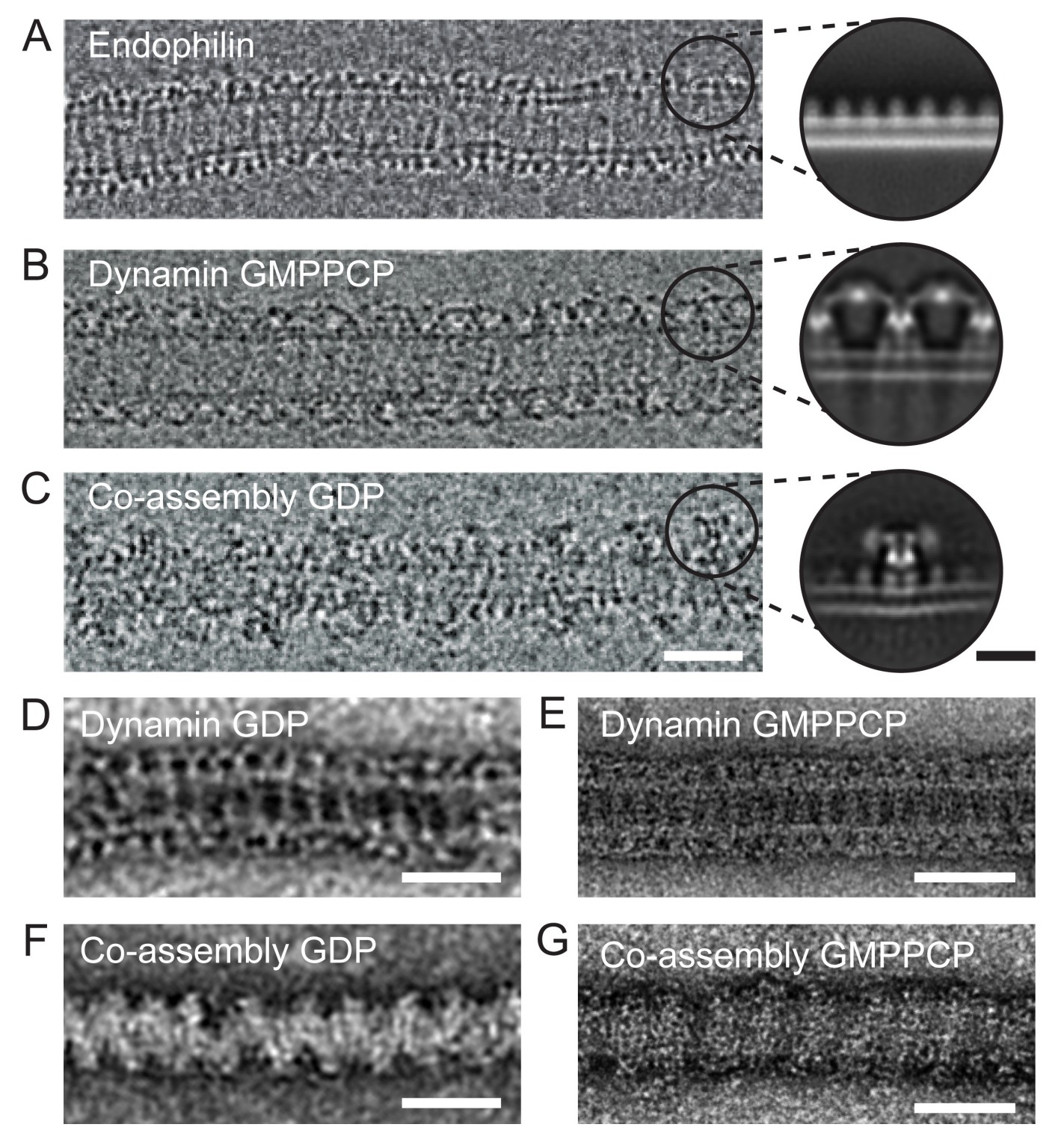

**Figure 3.** Endophilin can interleave between turns of a dynamin oligomer. CryoEM of coated tubules of (**A**) endophilin alone (**B**) dynamin (GMPPCP) or (**C**) the endophilin-dynamin co-complex (GDP). Black circles on the cryoEM micrographs (**A-C**) delineate example particle coordinates that were picked to generate the cryoEM 2D class averages to the right. Negative stain micrographs of protein-coated tubules in GDP (**D, F**) and GMPPCP (**E, G**) nucleotide-bound states. (**D-E**) Dynamin only versus (**F-G**) endophilin-dynamin co-assembly. Black scale bar, 10 nm. White scale bars, 50 nm.

DOI: https://doi.org/10.7554/eLife.26856.008

The following figure supplement is available for figure 3:

*Figure 3 continued on next page*

*Figure 3 continued*

**Figure supplement 1.** 1-start and 2-start helices formed by Dynamin-1 with the non-hydrolyzable GTP analog GMPPCP.

DOI: https://doi.org/10.7554/eLife.26856.009

overexpressing endophilin-RFP. As SKMEL cells primarily express the ubiquitous isoform dynamin2, we overexpressed endophilinA2, the ubiquitous isoform of endophilin (endophilinA2-RFP plasmid kindly provided by Emmanuel Boucrot). Indeed, overexpression of endophilinA2-RFP in SKMEL-dynamin-GFP cells correlated with much brighter dynamin foci (*Figure 4A*, purple arrows) than in non-transfected cells (*Figure 4A*, green arrows). EndophilinA2-RFP colocalized with those bright dots. Overexpression of endophilinA2-RFP also delayed the kinetics of dynamin-GFP, as seen comparing kymographs taken in non-transfected cells (*Figure 4B*) with kymographs of cells overexpressing endophilinA2-RFP (*Figure 4C*). Similar delays were observed at single pit levels (*Figure 4D,E*). The average duration of dynamin2 foci was 13.3 ± 14.7 s in non-transfected cells and went above 100 s when endophilinA2-RFP was overexpressed (*Figure 4F*).

We then wondered if this fission block had any functional consequence on the endocytic uptake of cargoes, and studied the uptake of transferrin while overexpressing endophilin. Indeed, SKMEL cells overexpressing endophilinA2-RFP show a lower internalization of transferrin (*Figure 4G*). While quantifying this effect on a large number of cells, the effect was statistically significant (*Figure 4H*, compare 'NT' and 'tot'). However, plotting the transferrin intensity as a function of the endophilin overexpression in single cells showed a substantial decrease of transferrin uptake in the most overexpressing cells (*Figure 4I*). We thus plotted the average value of the transferrin fluorescence uptake in cells overexpressing endophilin-RFP with a lower value than 10,000 Fluorescence Arbitrary Units (F.A.U) (see *Figure 4H*,<10,000), or with a higher value than 10,000 F.A.U (*Figure 4H*,>10,000). While the uptake of transferrin was not significantly different from the non-transfected cells when endophilin fluorescence was less than 10,000 F.A.U, the transferrin uptake was significantly reduced in cells expressing endophilin above 10,000 F.A.U (*Figure 4H*). Thus, endophilin over-expression is inversely correlated to transferrin uptake in these cells.

As in our in vitro experiments, the ratio between dynamin and endophilin was a critical parameter to observe the fission inhibition, regardless of the absolute amount of the proteins. We thus tested the possibility that the inhibition of dynamin-mediated fission could be rescued in cells overexpressing endophilin by simply co-overexpressing dynamin. Indeed, when dynamin2-cherry and endophilinA2-GFP were co-overexpressed in SKMEL dynamin2-GFP cells, no significant inhibition of the transferrin uptake was observed (*Figure 4H*, Dyn2 OE). From these in vivo results, we concluded that co-assembly with endophilin can inhibit dynamin-mediated fission in a concentration-dependent manner.

## Discussion

In this study, we have tested the effect of endophilin on dynamin's membrane fission activity. Our results show that the non-physiological 4x molar excess of endophilin inhibits fission completely by blocking dynamin constriction. We measured fission rates and efficiencies for different endophilin-dynamin molar ratios in a cell-free system and concluded that the inhibition was not only a time delay but also a decrease in fission efficiency up to a complete blockage. Deletion construct experiments indicated that both endophilin's membrane binding N-BAR domain and direct dynamin binding SH3 domain contributed to full inhibition.

We have used negative stain EM and cryoEM to visualize the endophilin-dynamin membrane bound co-complex and found that turns of endophilin intercalate between turns of dynamin polymers (*Figure 3*, model in *Figure 4J*). This structural arrangement explains the apparent increase in dynamin pitch reported earlier on blocked and elongated vesicle necks or tubular membranes (*Iversen et al., 2003*; *Takei et al., 1998*; *Takei et al., 1995*). Interestingly, all known BAR, N-BAR and F-BAR partners of dynamin were shown to change the architecture of the dynamin coat in similar ways when co-assembled on membrane tubules (*Itoh et al., 2005*). This observation and our results suggest that many BAR family proteins with dynamin-binding SH3 domains can potentially inhibit membrane fission by co-assembling with dynamin polymers when present in stoichiometric excess relative to dynamin.

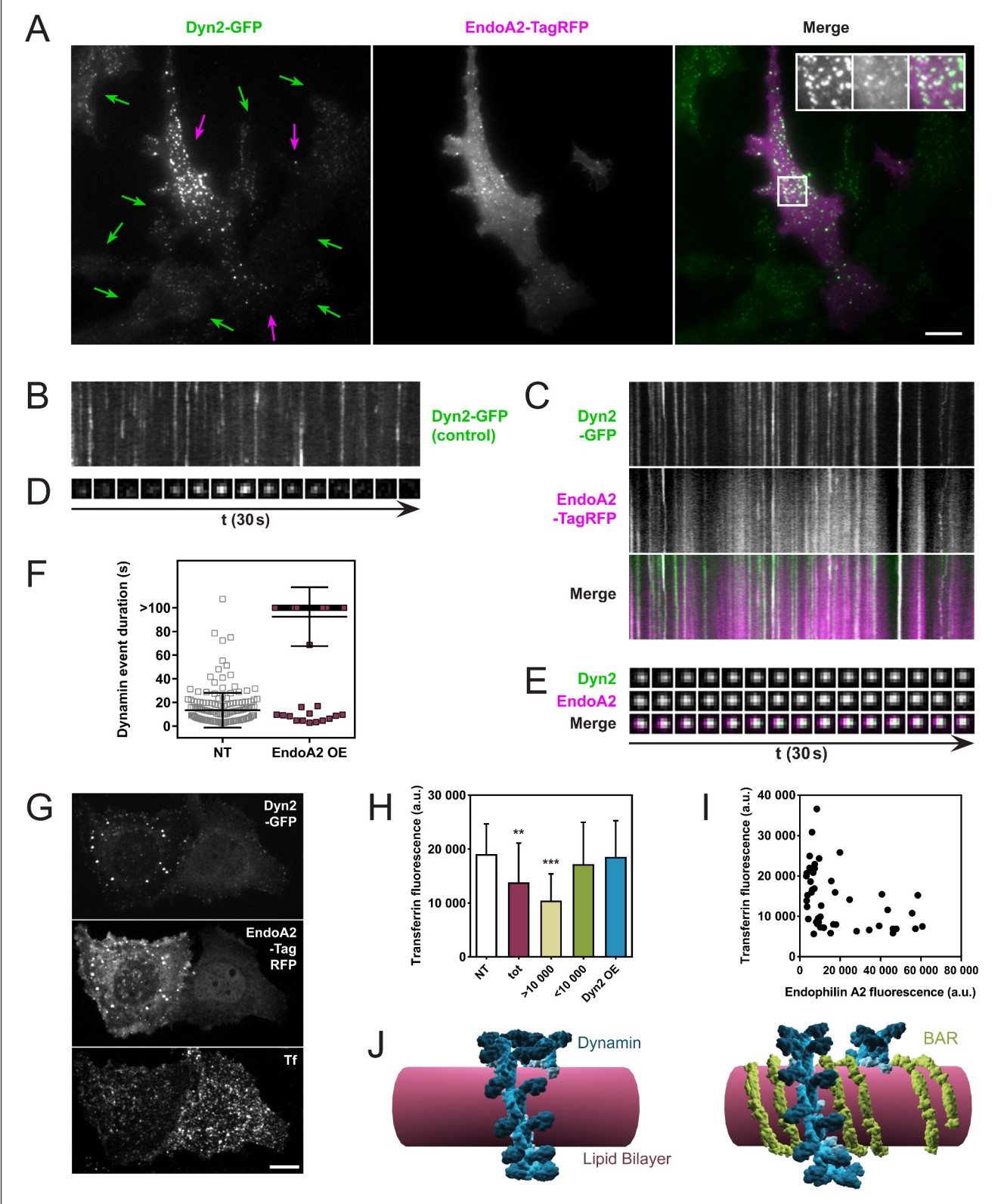

**Figure 4.** Endophilin overexpression blocks endocytic pits in vivo. (**A**) TIRFM images of genome-edited SK-MEL-2 cells overexpressing endophilinA2-TagRFP. Purple arrows, transfected cells; green arrows, non-transfected cells. Inset: Magnification of white box; left, Dyn2-GFP signal; middle, EndoA2-TagRFP signal; right, merge. (**B, C**) Kymographs of dynamin2-GFP foci from non-transfected and transfected cells (respectively) show delayed kinetics of dynamin foci in cells overexpressing endophilinA2-TagRFP. Length of kymographs, 100 s. (**D, E**) Montage of representative dynamin foci from non-

*Figure 4 continued on next page*

Figure 4 continued

transfected and transfected cells (respectively). (F) Distribution of dynamin event durations for non-transfected (NT) and endophilinA2-TagRFP overexpressing cells. N(NT)=184, n(EndoA2 OE)=177. Source data are available in the *Figure 4—source data 1*. (G) Confocal images of SK-MEL-2 cells expressing different levels of endophilinA2-TagRFP. (H) Quantification of transferrin fluorescence in non-transfected cells (NT), cells overexpressing endophilinA2-TagRFP (tot), cells with high (>10 000) and low (<10 000) endophilinA2-TagRFP levels, and cells co-overexpressing dynamin2 and endophilinA2-GFP (Dyn2 OE). n > 20 for all conditions. Source data are available in the *Figure 4—source data 2*. (I) Plot of transferrin fluorescence signal vs. endophilinA2-TagRFP fluorescence from the tot condition in (H). (J) Model for inhibition of dynamin constriction by BAR domain proteins. G domains (dark blue) of adjacent dynamin rings interact and drive conformational changes to constrict the underlying membrane tube. When BAR proteins such as endophilin (green) are present between the dynamin rings, the G domains of dynamin cannot interact anymore. Therefore, constriction and following fission are inhibited. Scale bars, 10 μm.

DOI: https://doi.org/10.7554/eLife.26856.010

The following source data is available for figure 4:

**Source data 1.** Dynamin event durations for non-transfected (NT) and endophilinA2-TagRFP overexpressing cells.

DOI: https://doi.org/10.7554/eLife.26856.011

**Source data 2.** Quantification of transferrin fluorescence in non-transfected cells, cells overexpressing endophilinA2-TagRFP, cells with high (>10,000) and low (<10,000) endophilinA2-TagRFP levels, and cells co-overexpressing dynamin2 and endophilinA2-GFP.

DOI: https://doi.org/10.7554/eLife.26856.012

The role of endophilins in membrane fission has already been extensively investigated. While the N-terminal amphipathic helices have been shown to promote fission by inducing very high curvature, the crescent shape of the BAR domain counteracted this effect by fixing membrane curvature above the spontaneous fission threshold (*Boucrot et al., 2012*). But endophilinA2 was recently shown to be needed for fission in clathrin-independent endocytic pathways specific to G-coupled receptors (*Boucrot et al., 2015*), or to Shiga-toxin (*Renard et al., 2015*). EndophilinA2 polymerizes into rigid scaffolds around preformed membrane tubules (*Simunovic et al., 2016*), which block diffusion of lipids when the membrane tube is under tension, driving thinning and fission of the tube (*Renard et al., 2015*; *Simunovic et al., 2017*). In contrast, we show here that overexpression of endophilin A2 inhibits fission at clathrin-coated pits, where dynamin is the primary fission machinery (*Figure 4*), which is consistent with the inhibitory action of endophilin on dynamin-dependent fission in our in vitro studies. While the physiological relevance of this inhibitory action may be questioned, as it only happens when endophilin is in excess, it nevertheless provides insight into the interplay of endophilin and dynamin.

Endophilin participates in the recruitment of dynamin to vesicle necks pointing to a positive role in promoting fission (*Farsad et al., 2001*; *Gad et al., 2000*; *Shupliakov et al., 1997*; *Sundborger et al., 2011*; *Wigge et al., 1997*). Additionally, lack of all three endophilins results in delayed endocytosis at synapses (*Milosevic et al., 2011*). One potential explanation for the conflict between this positive role in dynamin's action and the inhibitory role demonstrated by the present study, is that dynamin is assembled initially as a constriction-inhibited co-polymer (*Wu et al., 2010*) at clathrin-coated pits. Endophilin and the other endocytic proteins may prepare the vesicle neck for fission by narrowing the neck and recruiting dynamin. However, the presence of BAR domain-containing proteins will also create a fission-inhibited zone of the tubule that may serve to position the dynamin-mediated fission reaction toward the junction between the cylindrical neck and the spherical vesicle. Further assembly of the dynamin polymer alone beyond the BAR-domain/dynamin co-assembly would create a fission competent dynamin collar closer to the vesicle.

While the above hypothesis is speculative, the physiological role of BIN1 (the muscle isoform of amphiphysin) and dynamin2 in T-tubule biogenesis may reveal the importance of the structural inhibition highlighted in our study (*Hohendahl et al., 2016*). In T-tubule biogenesis, both mutants of BIN1 and dynamin cause centro-nuclear myopathies (CNMs), with a strong defect in T-tubule morphology (*Bitoun et al., 2005*; *Nicot et al., 2007*). BIN1 and dynamin2 have been directly implicated in the formation of T-tubules formed from the plasma membrane of myocytes (*Lee et al., 2002*). Strikingly, dynamin mutants with muscle-specific phenotype have increased fission capacities, causing T-tubules fragmentation (*Hohendahl et al., 2016*). BIN1 mutants disrupt membrane and/or dynamin binding, but display the same fragmented T-tubule phenotype. These results suggest that the co-assembly of dynamin and BIN1 along T-tubules is a physiological example of a fission-

incompetent co-assembly, providing a role for BIN1 not just in shaping the T-tubule but for structural inhibition of dynamin-mediated membrane fission.

## Materials and methods

### Materials

Guanosine 5′-triphosphate (GTP, 10106399001) was purchased from Roche Diagnostics GmBH, Mannheim, Germany. Guanosine 5′-diphosphate (GDP, NU-1172) was purchased from Jena Bioscience GmBH, Jena, Germany. β,γ-Methyleneguanosine 5′-triphosphate sodium salt (GMPPCP, M3509) was purchased from Sigma-Aldrich, St. Louis, USA. Glutaraldehyde, 25% EM grade (111-30-8) was purchased from Polysciences, Inc., Warrington, USA. Casein (β-Casein from bovine milk bioultra, C6905) and NaCl were purchased from Sigma-Aldrich Chemie GmBH, Buchs, Switzerland. Dithiothreitol (DTT), ethylenediaminetetraacetic acid (EDTA), ethylene glycol-bis(β-aminoethyl ether)-N,N,N′,N′-tetraacetic acid (EGTA), 4-(2-hydroxyethyl)−1-piperazineethanesulfonic acid (HEPES), isopropyl β-D-1-thiogalactopyranoside (IPTG), piperazine-N,N′-bis(2-ethanesulfonic acid) (PIPES), sucrose, tris (hydroxymethyl)aminomethane (Tris) and Triton X-100 were purchased from AppliChem GmBH, Darmstadt, Germany. $MgCl_2$ was purchased from Acros Organics, New Jersey, USA. Brain polar lipids (BPL, Brain Polar Lipid Extract), Brain L-α-phosphatidylethanolamine (Brain PE), Brain L-α-phosphatidylinositol-4,5-bisphosphate (Brain $PIP_2$), Cholesterol, 1,2-dioleoyl-sn-glycero-3-phosphocholine (DOPC), 1,2-dioleoyl-sn-glycero-3-phosphoethanolamine (DOPE), 1,2-dioleoyl-sn-glycero-3-phospho-L-serine (DOPS), 1,2-distearoyl-sn-glycero-3-phosphoethanolamine-N-[biotinyl(polyethylene glycol)−2000] (DSPE-PEG2000 Biotin), Egg L-α-phosphatidylcholine (EPC), Liver L-α-phosphatidylinositol (Liver PI), 1-palmitoyl-2-oleoyl-sn-glycero-3-phosphocholine (POPC) and 1-palmitoyl-2-oleoyl-sn-glycero-3-phospho-L-serine (POPS) were purchased from Avanti Polar Lipids, Alabaster, USA. BODIPY TMR-PtdIns(4,5)$P_2$ (TMR-$PIP_2$) was purchased from Echelon Biosciences, Salt Lake City, USA. Brain Extract form bovine brain, Type I, Folch Fraction I (Folch) was purchased from Sigma-Aldrich, St. Louis, USA. COOH-coated beads (120 nm or 320 nm diameter, PC02N) were purchased from Bangs Laboratories, Fishers, USA. Streptavidin-coated beads (SVP-30–5) were purchased from Spherotech, Lake Forest, USA.

### Dynamin purification

Human dynamin1 was purified as described from baculovirus-infected Sf9 cells expressing recombinant dynamin1, via affinity purification with the SH3 domain of rat amphiphysin1 (*Kalia et al., 2015*; *Stowell et al., 1999*).

Typically, 2 l of Sf9 cells were infected with recombinant baculovirus and incubated under agitation for 3 days. The cell lysate was resuspended in 50 ml of buffer A (20 mM HEPES, 100 mM NaCl, 1 mM EGTA, 1 mM DTT, pH 7.4)+protease inhibitor tablets (cOmplete EDTA-free Protease Inhibitor Cocktail, 05056489001, Roche Diagnostics GmbH, Mannheim, Germany). After homogenization and centrifugation, the supernatant was incubated under agitation for 2 hr at 4°C with 5 ml glutathione Sepharose beads (Glutathione Sepharose 4B, 17-0756-05, GE Healthcare Bio-Sciences AB, Uppsala, Sweden) to which 10 mg of GST-tagged amphiphysin1-SH3 domain had been attached. After three washes with buffer A, the bead suspension was loaded into a column and the protein eluted with high-salt buffer B (20 mM PIPES, 1.2 M NaCl, 1 mM DTT, pH 6.5). The protein-positive fractions were pooled, dialyzed against GTPase buffer (20 mM HEPES, 100 mM NaCl, 1 mM $MgCl_2$, pH 7.4) and concentrated on an Amicon 50 kDa column (Merck Millipore Ltd., Carrigtwohill, Ireland).

### Purification of endophilin constructs

Rat endophilinA1 was produced from a pGEX-6P vector in *E. coli* upon induction with IPTG for 3 hr at 37°C. The pellet was resuspended in PBS + 1% Triton X-100 + cOmplete protease inhibitor tablets (see above). The 50 ml supernatant after sonication and centrifugation was incubated for 1 hr at 4°C with 7.5 ml glutathione Sepharose beads (ABT, Miami, USA). Following two washes with cleavage buffer (50 mM Tris, 150 mM NaCl, 1 mM EDTA, 1 mM DTT, pH 7.5), PreScission enzyme was added to cut off the endophilin from the column during an overnight incubation at 4°C with agitation. Protein positive fractions of the flow through were pooled, concentrated on a Vivaspin 6 10 kDa column (Sartorius Stedim Biotech GmbH, Göttingen, Germany) and dialyzed against GTPase buffer.

The BAR domain was purified like full-length endophilin, apart from the PreScission treatment which was done in solution, prior to adding the suspension to a column. The SH3 domain was purified similarly to full-length endophilin, but with the following differences: induction of protein production with IPTG was performed only during 2 hr; the SH3 domain was eluted from the beads on a column by addition of glutathione; only then the GST-tag was cut off from the SH3 domain by PreScission treatment; a second incubation with glutathione Sepharose beads followed to remove free GST and PreScission which also contained a GST-tag.

## Protein labeling

Fluorescent dynamin was obtained by labeling with Atto 488 iodoacetamide (AD 488–111, ATTO-TEC GmbH, Siegen, Germany) at a molar ratio of 1:3 (protein vs. label), following the manufacturer's labelling protocol. Half fluorescently labeled and half non-labeled dynamin was used in fluorescence microscopy. Endophilin and endophilin BAR domain were labeled with Atto 390 NHS-ester (AD 390–31, see above) at a molar ratio of 1:3. In some cases, endophilin was labeled with Atto 390 iodoacetamide (AD 390–111) or Atto 565 iodoacetamide (AD 565–111; membrane sheets colocalization).

## Vesicle preparation

Three different membrane compositions were used for unilamellar vesicles (GUVs) in the tube-pulling experiments: composition 3321 (*Figure 2*): 28.5% DOPC, 28.5% DOPE, 20% DOPS, 9% Brain $PIP_2$, 13% cholesterol, 1% TMR-$PIP_2$; composition 80:20 (*Figure 3*): 80% EPC +19% $PIP_2$ +1% TMR-$PIP_2$; composition 3312 (*Figure 3D–I*): 29% DOPC, 29% DOPE, 11% DOPS, 17% Brain $PIP_2$, 13% cholesterol, 1% TMR-$PIP_2$. Percentages are mole percent, each of these compositions were supplemented with 0.03% DSPE-PEG(2000) Biotin to allow binding to the streptavidin-coated bead.

For preparing GUVs, 10–15 µl lipid mix of 1–2 g/l lipids in pure chloroform were deposited on two indium tin oxide (ITO)-coated slides and dried for 1 hr at 55°C. A chamber was assembled with a teflon O-ring between the two slides and it was filled with ~400 µl sucrose of 200–230 mOsm (adjusted ± 1 mOsm to the respective experimental buffer). The GUVs were electroformed by applying sinus voltage of 1 V at 10 Hz for 1 hr at 55°C.

SUVs for electron microscopy were prepared by premixing lipid stocks stored in chloroform into the correct molar ratios. The mixtures were then placed into a 5 ml reaction vial (TS-13223, Thermo Fisher Scientific, Waltham, MA, USA) and the lipids were dried onto the glass surface by vortexing over a stream of nitrogen gas. Lipids were then reconstituted in 0.2 ml hexane, followed by the same vortex drying procedure, to obtain an evenly dried lipid coat on the reaction vial. The samples were lyophilized for >1 hr to remove any residual solvent. Lipid mixtures were then reconstituted in GTPase buffer to make a final SUV solution of 1 g/l. The sample mixed 12–16 hr in a tube rotator to fully reconstitute the lipids from the side of the glass reaction vial. The samples were then subjected to five freeze thaw cycles to burst any multi-lamellar liposomes. SUV compositions were either (1) DOPS liposomes (*Figure 3A,D–G*): 28% DOPC, 28% DOPE, 12% DOPS, 17% Brain $PIP_2$, and 15% cholesterol or (2) 100% DOPS liposomes (*Figure 3B–C*). Percentages are mole percent.

SUVs for the GTPase assay were made by drying 200 µl 10 g/l chloroform solution of 95% BPL and 5% Brain $PIP_2$ in a glass vial with a nitrogen flow. The lipids were rehydrated in 1 ml GTPase buffer and incubated at 37°C for 1 hr, followed by vortexing and sonication.

## Membrane sheets assay

A lipid solution of 10 mg/ml containing 26% Brain PE, 4.5% Liver PI, 26% POPS, 30.5% POPC and 13% cholesterol was prepared, which was then supplemented with 5% final Brain $PIP_2$. Glass coverslips were cleaned with chloroform. Droplets of 1.5 µl of above lipid solution were deposited on the coverslip, allowed to dry, and then dried in the vacuum oven for at least 1 hr. A chamber of ~25 µl was built by placing the coverslip onto a glass slide, the dried lipid drops facing the glass slide, using a double-sided Scotch (3M) tape as a spacer (see *Figure 2* in [*Itoh et al., 2005*]). The lipids were fully re-hydrated by injecting 25 µl of 4 g/l casein in GTPase buffer. Rehydration generated membrane sheets, as previously described (*Suzuki and Masuhara, 2005*). The glass slide was placed on the stage of either an Axiovert 200 ZEISS (Germany) microscope for observation with a JAI Pulnix (USA) TM1400CL camera and DVR software (Advanced Digital Vision Inc. USA), or an inverted spinning

disk confocal microscope (Nikon Eclipse T1 with spinning disk from 3I, Denver, CO, USA) with a 100x oil objective (Nikon CFI Apo TIRF 100x NA 1.4).

To generate tubules coated with both endophilin and dynamin, endophilin was added to the chamber with a molar excess of at least four times compared to dynamin (typically 16 μM). Then, 5 μl of a dynamin containing solution (typically 4 μM) were applied to one side of the chamber and the deformation of membrane sheets produced by its diffusion into the chamber under Differential Interference Contrast (DIC) settings. 5 μl of 1 mM GTP containing buffer were added after formation of the tubules.

For experiments involving COOH-coated beads (negatively charged at neutral pH, 320 nm diameter, Bangs Labs, USA), beads were diluted 500-1000x in the dynamin solution prior to injection. Only tubules not adherent to the glass surface throughout their length were selected for observation.

### Bead rotation analysis

For the analysis of the bead movement, movies were contrasted using virtualdub, transformed to eight bit grayscale stack (.stk) files using the ImageJ software, and the spinning beads were tracked using the Tracking Function of the Metamorph software (Molecular Devices Corp., USA). For each tubule type, the average maximal angular speed was calculated from the maximal speeds (which in turn was an average of at least three turns) of 15 to 30 beads.

### Tube pulling assay

A ~2 mm high chamber was assembled between two rectangular horizontal glass slides and mounted on the stage of an inverted scanning confocal microscope (Eclipse T1 with A1R scanner, Nikon, Tokyo, Japan) with a 100x oil objective (CFI Plan Apo VC 100X Oil NA 1.4, Nikon, see above). The microscopy setup included home-made optical tweezers, for which an ytterbium fibre laser (IPG Photonics, Oxford, USA) was focused through the microscopy objective. The chamber was pretreated with 4 g/l casein in GTPase buffer and then filled by capillarity with a mixture of GTPase buffer and (0.02% (v/v)) streptavidin-coated beads of 3.05 μm average diameter. A GUV was aspirated by a glass pipette (*Figure 2A*) which was controlled by a micromanipulator (MP-225, Sutter Instrument, Novato, USA). A streptavidin coated bead was held by the optical tweezers, and a nanotube pulled from this bead by first applying it onto the GUV to induce adhesion, and then pulling the GUV away. Time-lapse imaging was started and the protein/GTP mixture injected with a second micropipette controlled by a manual micromanipulator (Narishige, Tokyo, Japan). In some experiments, a third micropipette was used for injection of a different protein mixture, controlled by a third micromanipulator (MP-285, Sutter, see above).

### GTPase assay

We used the malachite green assay to monitor the release of inorganic phosphate due to the GTPase activity of dynamin (*Quan and Robinson, 2005*). GTPase activity was measured in the presence of SUVs containing 95% BPL and 5% Brain PIP$_2$ (see vesicle preparation section for details). All mixes were performed on the same 96-well plate, and a molar ratio of four between endophilin and dynamin was used. The assay was performed in a total reaction volume of 50 μl, containing typically 200 nM dynamin (plus eventually 800 nM endophilin), 2 μl SUVs (0.4 g/l final concentration) and 100 μM GTP. The buffer of the reaction was GTPase buffer. Incubation prior to addition of malachite working solution was for 30 min at 37°C.

### Image analysis

Image analysis was performed with ImageJ and data fitting with Matlab (*Source code files 1* and *2*). Dynamin event analysis was performed with the detection algorithm in the program Utrack (*Jaqaman et al., 2008*) and Matlab code (PostUtrack_1 and postUtrack_2, manuscript in revision in eLife) developed by Rafael Sebastian (University of Valencia, Spain).

### Membrane remodeling reactions for electron microscopy

Purified protein reagents and unilamellar liposomes were mixed together and allowed to incubate at room temperature for 30 min. To generate the homo- or hetero-polymers of dynamin and

endophilin, protein was added to a final concentration of 4–20 µM and mixed with the appropriate nucleotide at final concentrations of 1–2 mM in GTPase buffer described above. The protein to lipid ratio was 0.5–1.0:2.0 by mass. Optimized reactions for cryoEM analysis were as follows. Endophilin only: 11 µM protein reacted with DOPS liposomes (0.5 g/l final concentration) for 30 min at room temperature. Dynamin only: 8 µM protein with 2 mM GMPPCP reacted with 100% DOPS liposomes (0.75 g/l final concentration) for 30 min room temperature. Co-complex: 6 µM dynamin and 6 µM endophilin with 2 mM GDP reacted with 100% DOPS liposomes (0.63 g/l final concentration) for 30 min room temperature. The complex reactions were then pelleted at 2152 relative centrifugal force (RCF) in a benchtop centrifuge for 5 min and 80% of the supernatant solution was removed from the tube. The remaining concentrated membrane-bound tubules were resuspended via pipetting and used for preparing co-complex samples.

## Negative stain electron microscopy

Protein remodeled membranes were prepared for TEM following established procedures (*Booth et al., 2011*). Continuous carbon grids (400 mesh copper) were prepared by glow-discharging the surface (PELCO EasiGlow, 15 mA, 0.39 mBar, 30 s). Each sample (5 µl) was allowed to absorb onto the grid surface for 2 min followed by blotting onto filter paper. The grids were then stained with 0.75% (w/v) uranyl formate for 30 s, blotted, and allowed to air dry. Samples were imaged with a Tecnai T12 microscope (FEI Company, Hillsboro, USA) equipped with a LaB6 filament and operated at 120 kV and data captured with a Gatan Ultrascan CCD camera (Gatan, Inc., Pleasanton, USA).

## Electron cryo-microscopy

Membrane remodeled protein reactions were applied to a glow-discharged (PELCO EasiGlow, 15 mA, 0.39 mBar, 30 s) Quantifoil holey carbon grid (R2/2, 200 Cu mesh or R1.2/R1.3, 200 Cu mesh) in a Vitrobot Mark III (FEI Company, Hillsboro, USA). Specifically, 3.5 µL of sample was applied in a 100% humidity 19°C chamber, allowed to absorb to the grid surface for 30 s, blotted against filter paper for 1.5–4.5 s (0 mm offset), and then plunge frozen into liquid ethane for the endophilin only and dynamin GMPPCP only samples. For the co-complex GDP samples, 3.0 µL of concentrated membrane-bound sample was applied to the EM grid in a 100% humidity 19°C chamber and allowed to incubate on the grid for 30 s. Then 0.5 µL of a 1% (v/v) glutaraldehyde solution (0.14% (v/v) final) was added to the droplet on the grid for 10 s. The sample was blotted against filter paper for 1.5–4.5 s (0 mm offset), and plunge frozen into liquid ethane. Samples were stored in liquid nitrogen until imaging could take place (*Grassucci et al., 2007*). Three datasets were collected corresponding to three different membrane remodeled protein samples: (1) endophilin only, (2) dynamin GMPPCP, (3) endophilin-dynamin complex GDP crosslinked with glutaraldehyde. Electron cryo-micrographs for all samples were collected using standard low-dose procedures through UCSFImage4 or SerialEM on a Tecnai TF30 Polara microscope (FEI Company, Hillsboro, USA) operated at 300 kV (*Li et al., 2015*). Images were collected at a nominal magnification of 31,000 x (endophilin and co-complex GDP) or 15,500 x (dynamin GMPPCP) with a K2-summit direct electron detector (Gatan, Inc., Pleasanton, USA) operating in super resolution mode corresponding to a 0.6078 or 1.245 Å/pixel pixel size, respectively. Micrographs for endophilin only were recorded with 6.0 s exposures (0.2 s frame rate) corresponding to a 30 frame image stack with a total dose of ~40 e⁻/Å² ($e^-/\text{Å}^2$). Motion correction was completed with UCSF MotionCorr throwing the first two frames and binning the micrographs by a factor of two (1.2156 Å/pixel) (*Li et al., 2013*). Electron cryo-micrographs for dynamin GMPPCP and co-complex GDP datasets were recorded with 8.0 s exposures (0.2 s frame rate) corresponding to a 40-frame image stack with total doses of ~20 e⁻/Å² ($e^-/\text{Å}^2$) and ~40 e⁻/Å² ($e^-/\text{Å}^2$). Motion correction for these two datasets was completed with UCSF MotionCor2 throwing the first frame (*Zheng et al., 2017*). The data collection parameters for all of the cryoEM datasets are summarized in *Table 1*.

## Electron cryo-microscopy image processing

Contrast transfer function (CTF) parameters were estimated using CTFFIND4 (*Rohou and Grigorieff, 2015*) or GCTF (*Zhang, 2016*). Filamentous assemblies were manually segmented using Relion2.0 helix manual picking functions. Linear segments defined by the lipid bilayer were chosen for each side of the protein coated lipid tubule. Extraction was performed by segmenting overlapping

**Table 1.** Data collection parameters

| Dataset | Endophilin | Dynamin GMPPCP | Co-complex GDP |
|---|---|---|---|
| Microscope | TF30 Polara | TF30 Polara | TF30 Polara |
| Detector | K2 Summit | K2 Summit | K2 Summit |
| Collection | UCSFimage4 | SerialEM | SerialEM |
| Pixel size (Å) | 1.22 | 2.49 | 1.22 |
| Exposure (sec) | 6.0 | 8.0 | 8.0 |
| Total Dose (e⁻/Å²) | 40 | 20 | 40 |
| Micrographs | 204 | 2006 | 1660 |
| Motion Correction | MotionCorr | MotionCor2 | MotionCor2 |
| Defocus Range (µm) | 0.6–2.6 | 0.3–5.0 | 0.8–2.8 |
| Particles contributing to class averages in *Figure 3* | 2047 | 2766 | 19,662 |

DOI: https://doi.org/10.7554/eLife.26856.013

segments (50 Å apart) for dynamin GMPPCP and co-complex GDP datasets. The endophilin only dataset was extracted into overlapping segments (41.3 Å). Image and processing parameters are summarized in *Table 1* for all three datasets. These extracted particles were then subjected to reference-free 2D classification through RELION2.0 (*Kimanius et al., 2016*; *Scheres, 2012*). Multiple rounds of classification were performed to identify structurally uniform classes that clearly identified the lipid bilayer and protein coats. Measurements of stalk-to-stalk distances on the 2D class averages for the dynamin only GMPPCP sample were carried out using ImageJ (*Figure 3—figure supplement 1*). Specifically, line profiles (grey scale pixel intensity versus distance) were generated and the distance between the pixels corresponding to adjacent stalks were calculated. For the co-assembly GDP dataset, distances between the dynamin stalks (along the tangent of the tubule, for both sides) were recorded from raw cryoEM micrographs (*Figure 3C*). The gaps between the adjacent dynamin stalks varied image to image, and the mean distance (± standard deviation) between adjacent stalks was calculated individually for n = 50 randomly chosen tubes.

## Cell transfection

Genome-edited SK-MEL-2 cells expressing the endogenous dynamin2 with a GFP tag (kind gift from David Drubin lab, UC Berkeley, USA; [*Doyon et al., 2011*]) were transfected using 0.5 µg of plasmids containing human endophilin A2 with C-terminal TagRFP-T under the control of the CMV promoter with X-tremeGENE HP DNA Transfection Reagent (06366236001, Roche Diagnostics GmbH, Mannheim, Germany) according to the manufacturer's protocol. The endophilin plasmid was kindly provided by Emmanuel Boucrot, University College London, UK. For the recovery experiments, human dynamin2 cherry (kindly provided by Christien Merrifield, LEBS, Gif-Sur-Yvette, France) was overexpressed together with a GFP version of human endophilinA2 (kindly provided by Emmanuel Boucrot [*Boucrot et al., 2015*]). Images of the cells were acquired on a microscope (Nikon Eclipse T1, Nikon Tokyo, Japan) with a 100x oil objective (Nikon CFI Apo TIRF 100x NA 1.4) using total internal reflection fluorescence (TIRF).

## Transferrin uptake

Cells were incubated with Alexa Fluor 647-conjugated transferrin (T23366, Life Technologies, distributed by ThermoFisher Scientific, Waltham, USA) for 15 min, washed with PBS and fixed with paraformaldehyde (PFA). Images were acquired on a confocal microscope (Nikon Eclipse T1, Nikon Tokyo, Japan) with spinning disk (3I, Denver, CO, USA) with a 100x oil objective (Nikon CFI Apo TIRF 100x NA 1.4). z-stack images in a total range of 10 µm were acquired and maximum projections obtained with ImageJ. The quantification of the fluorescence signal was performed in ImageJ by measuring the mean grey value for the transferrin channel (far red) per pixel in the area of the cells.

## Acknowledgements

The authors thank Emmanuel Boucrot, David Drubin and Christien Merrifield for the materials provided. The authors thank Janet Iwasa for the assistance and training for generating the model figure with Autodesk Maya. They also thank Rafael Sebastian (University of Valencia) for the software postUtrack_1 and postUtrack_2 (part of manuscript in revision in eLife). We thank Michael Braunfeld, David Bulkley, and Alexander Myasnikov of the UCSF Center for Advanced cryoEM, which is supported in part from NIH grants S10OD020054 and 1S10OD021741 and the Howard Hughes Medical Institute. We also thank the QB3 shared cluster and NIH grant 1S10OD021596-01 for computational support and David Belnap for electron microscopy training (University of Utah). AF acknowledges funding by a Faculty Scholar grant from the Howard Hughes Medical Institute, the Searle Scholars Program, NIH grant 1DP2GM110772-01, and the Chan Zuckerberg Biohub. AR acknowledges funding from Human Frontier Science Program CDA-0061–08, the Swiss National Fund for Research Grants N°31003A_130520 and N°31003A_149975, and the European Research Council Starting Grant N° 311536 (2011 call). AR and VG acknowledge funding from the Initial-Training Network TRANSPOL (Marie Curie action grant #264399). PDC acknowledges funding from NIH grant NS036251 and the Howard Hughes Medical Institute.

## Additional information

### Funding

| Funder | Grant reference number | Author |
|---|---|---|
| H2020 European Research Council | 311536 | Aurélien Roux |
| Schweizerischer Nationalfonds zur Förderung der Wissenschaftlichen Forschung | 31003A_130520 | Aurélien Roux |
| Human Frontier Science Program | CDA-0061-08 | Aurélien Roux |
| National Institutes of Health | 1DP2GM110772-01 | Adam Frost |
| Schweizerischer Nationalfonds zur Förderung der Wissenschaftlichen Forschung | 31003A_149975 | Aurélien Roux |
| Howard Hughes Medical Institute | 55108523 | Pietro De Camilli |
| Searle scholars award | 13SSP218 | Adam Frost |
| National Institutes of Health | NS036251 | Pietro De Camilli |
| TRANSPOL | Marie Curie action grant #264399 | Valentina Galli Aurélien Roux |

The funders had no role in study design, data collection and interpretation, or the decision to submit the work for publication.

### Author contributions

Annika Hohendahl, Nathaniel Talledge, Conceptualization, Data curation, Formal analysis, Investigation, Methodology, Writing—original draft; Valentina Galli, Conceptualization, Data curation, Investigation; Peter S Shen, Data curation, Investigation, Methodology; Frédéric Humbert, Methodology; Pietro De Camilli, Conceptualization, Supervision, Funding acquisition; Adam Frost, Conceptualization, Data curation, Supervision, Investigation, Visualization, Methodology, Writing—original draft, Project administration; Aurélien Roux, Conceptualization, Supervision, Funding acquisition, Investigation, Visualization, Methodology, Writing—original draft, Project administration

### Author ORCIDs

Annika Hohendahl https://orcid.org/0000-0002-0852-1826
Nathaniel Talledge http://orcid.org/0000-0002-4603-0120

Adam Frost https://orcid.org/0000-0003-2231-2577
Aurélien Roux http://orcid.org/0000-0002-6088-0711

**Decision letter and Author response**
Decision letter https://doi.org/10.7554/eLife.26856.017
Author response https://doi.org/10.7554/eLife.26856.018

## Additional files

**Supplementary files**

• Source code file 1. Matlab code for fitting of cumulative fission probability data for *Figure 2E and H*. The code determines the variables a and τ based on the experimental data, for a fit to a*(1-exp(-t/τ)).
DOI: https://doi.org/10.7554/eLife.26856.014

• Source code file 2. Matlab code for fitting of cumulative fission probability data for *Figure 2L*. The code determines the variables a and τ based on the experimental data, for a fit to a*(1-exp(-t/τ)).
DOI: https://doi.org/10.7554/eLife.26856.015

• Transparent reporting form
DOI: https://doi.org/10.7554/eLife.26856.016

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
