## [Decision Letter]

Thank you for submitting your article "Structural relationship between dynamin and endophilin at the membrane interface" for consideration by *eLife*. Your article has been reviewed by three peer reviewers, and the evaluation has been overseen by a Reviewing Editor and Randy Schekman as the Senior Editor. The following individuals involved in review of your submission have agreed to reveal their identity: Volker Haucke (Reviewer #2); Anna Sundborger (Reviewer #3).

The reviewers have discussed the reviews with one another and the Reviewing Editor has drafted this decision to help you prepare a revised submission.

Summary:

There is a consensus among the reviewers that the in vivo and in vitro experiments clearly show excess endophilin blocks dynamin mediated fission when assembled together on lipid tubes and when overexpressed in cells. This is especially convincing in Figure 2 where they label for both endo and dynamin on the same tube. In these experiments both proteins are added simultaneously. However, in Figure 1 it is not clear if dynamin binds or how much binds to the preassembled endophilin tubes since here endophilin is added prior dynamin. Either fluorescently tagged protein or images of endophilin alone compared to endophilin and dynamin would help convince the readers. As suggested by one reviewer, the statement that the Bead experiments shows endophilin blocks dynamin mediated constriction should be rephrased to state that dynamin may constrict without causing bead rotation (supercoiling). The reviewers all agree that the in vivo assays in Figure 4 are convincing and clearly show overexpressing endophilin delays dynamin mediated fission and decreases transferrin uptake in proportion to the levels of endophilin expression.

The EM experiments have caused the most concern. It is unclear if the 2D class averages are from the cryo-EM data or negative stain. The Materials and methods suggest the 2D class averages are from cryo-EM data and if this is the case, stating this in the text and figure legends would validate the statement in the Abstract that cryo-EM was used to show the endophilin intercalates between turns of the dynamin helix. As commented by two of the reviewers, the resolution of the class averages is poor and more evidence is needed to support the model that endophilin prevents dynamin interactions in trans across the rungs of the helix. In addition, more quantification of the EM images would also help support the conclusions such as measurements between the GTPase domains in adjacent rungs, change in helical pitch and average diameter. A GTPase assay would help support the lack of G domain trans interactions.

The reviewers also agree that an expanded discussion of the results, taking into consideration other findings, would be beneficial. For example, how do the results shed light on the role of endophilin in the uptake of Shiga toxin, what is the role of endophilin's amphipathic helix, how does the data relate to physiological conditions and previously been published structural studies.

In addition, considering the main arguments supporting the conclusions are from cell biology and light microscopy results and not from structural studies, it is recommended that the title be reworded and not include the work 'Structural', which may be misleading and not reflecting the strengths of the paper.

Overall, the reviewers agree that the data is of high quality and value and by incorporating their suggestions, the manuscript will be of interest to a wide audience, especially the fields of endocytosis and membrane remodeling.

Essential revisions:

1) For the EM data, improve the resolution to support the conclusions and add quantification and measurements.

2) Expand the Discussion to incorporate other findings and further discuss potential mechanisms of action in vivo.

Reviewer #1:

Manuscript on membrane fission driven by dynamin. It is observed that endophilin inhibits dynamin-mediated tubule constriction and fission on model membranes by reducing fission rate and efficiency. Using cryo-EM, evidence is provided for an intercalation of endophilin between turns of dynamin oligomers. In cells, the overexpression of endophilin inhibits transferrin uptake.

The study presents a tour de force approach of modern membrane biophysics, including different model membrane systems, visual inspection of molecular dynamics at the nanoscale, optical tweezer setups, careful quantification, and some cryo-EM and cell biological experimentation. The body of data is quite convincing. The possibly biggest limitation comes from the fact that no evidence could be provided for this mechanism to operate under physiological conditions. The authors seem to be aware of this themselves, as they start the Discussion by stating that "the non-physiological 4x molar excess of endophilin inhibits scission completely by blocking dynamin constriction". They then suggest that "many BAR family proteins with dynamin binding SH3 domains can potentially inhibit membrane fission by co-assembling with dynamin polymers when present at a stoichiometric excess relative to dynamin". Any more direct evidence for this would clearly strengthen the physiological claim of this manuscript in a decisive manner.

Reviewer #2:

In this elegant study Hohendahl et al. have studied the debated role of the NBAR protein endophilin in dynamin-mediated fission in vitro and in living cells. They use membrane-sheet and GUV based assays to convincingly demonstrate that endophilin inhibits membrane fission of tubules coated with dynamin in a dose-dependent manner that requires its SH3 domain and, hence, association with dynamin. Further, negative stain EM analysis of lipid tubes coated with either dynamin or endophilin alone or both proteins together suggests that endophilin intercalates between dynamin-GDP rungs and thereby impairs fission triggered by adjacent GTPase domains that are predicted not to be able to interact in trans. Finally, experiments in genome engineered SKMEL cells expressing dynamin 2-GFP from its endogenous locus show that high level overexpression of endophilin A2 increases dynamin 2 lifetimes at CCPs and inhibits CME of transferrin, a phenotype rescued by co-overexpression of dynamin 2. These data are taken to suggest that endophilin and, possibly, other BAR domain proteins may create a fission-inhibited zone that could serve to restrict the dynamin-based fission reaction at the vesicle neck.

This is an interesting and scholarly study of high general interest and the data are of excellent quality. I, thus, have only a few relatively minor questions and suggestions for improvement.

1) My only point of concern pertains to the (central) proposal that endophilin assemblies intercalate between adjacent dynamin rungs in co-assembled polymers. Given the low resolution of the negative stain TEM I feel that additional evidence for this is required. What is the exact distance between GTPase domains in adjacent rungs? Can GTPase domains assembled on lipid tubes be crosslinked in the presence of endophilin?

If the model was correct then addition of endophilin should inhibit lipid-stimulated GTPase activity of dynamin in a dose-dependent manner. Has this been tested using the GUV system?

2) Does inhibition of dynamin-mediated fission depend on the amphipathic helices within endophilin A2?

3) How do these data fit with the recent proposal that endophilin facilitates fission, e.g. during CIE of Shiga toxin? This point at least should be discussed. In general I feel many readers would benefit from an extended discussion of the new findings and how they relate to previous studies in the literature.

Reviewer #3:

This study by Hohendahl et al. aims to resolve a hotly debated issue – do BAR proteins inhibit or promote dynamin-mediated fission? In a combination of in vitro fission assays, cryo-EM and live cell imaging, the authors show evidence of the BAR protein endophilin acting as an inhibitor of dynamin-mediated membrane fission.

In Figure 1 the authors use an in vitro assay to investigate the effect of endophilin on dynamin-mediated membrane tubulation and GTP-dependent fission. They convincingly show that addition of endophilin results in the formation of tubulates that withstand addition of GTP, suggesting that endophilin inhibits dynamin-mediated fission. They further investigate whether the presence of endophilin also interferes with GTP-hydrolysis-driven constriction of dynamin assayed by rotation of beads in a chamber. The presence of endophilin blocked dynamin-mediated bead rotation, suggesting that endophilin prevents not only fission, but also dynamin-mediated conformational changes. Comments, the authors claim that these results show that endophilin blocks dynamin constriction. However, the assay in question detects coiling of dynamin-decorated lipid tubes, rather than dynamin-mediated constriction. Coiling of dynamin-decorated lipid tubes may or may not have any significance to fission in vivo. Also, recent studies show that constriction of dynamin scaffolds occur upon GTP binding rather than hydrolysis (Sundborger et al., 2014). Therefore, these results should be regarded as evidence of endophilin blocking dynamin-mediated GTP-dependent conformational changes, but not necessarily membrane constriction, which is likely to occur simultaneous to dynamin scaffold assembly.

In Figure 2 the authors show inhibition of dynamin-mediated GTP-dependent fragmentation of nanotubes pulled from GUVs in the presence of GTP and endophilin. This inhibition is not observed in the presence of BAR domain only, indicating that the effect is dependent on SH3-PRD interactions between the two proteins. Comment: This assay fails to address the role of endophilin as a membrane remodeling protein. By using already constricted nanotubes, this assay bypasses a crucial step in the dynamin-mediated fission process – membrane constriction. Endophilin, the most likely candidate to promote constriction of the membrane upstream of dynamin recruitment, may regulate dynamin assembly as a function of curvature induction. This concept is even supported by previous work by the authors (Roux et al., 2010). Consequently, when using already-constricted tubes, the authors effectively bias their attempts to define the molecular mechanisms of endophilin-mediated inhibition of dynamin by not allowing it to act as a membrane remodeling protein, but rather as a potential recruiter of dynamin.

In Figure 3 the authors use cryo-EM to visualize the endophilin-dynamin complex assembled on lipid tubes. These results show the impact of endophilin on dynamin scaffold assembly. Comments: The 2D class averages presented in this figure in the key point of the paper and should be emphasized. This data is confirming previously suggested mechanisms of endophilin inhibiting fission by preventing proper dynamin assembly. The resolution of the images does not support such claims, which should be clearly stated in the text. Furthermore, there is no evidence that endophilin actually interleaves the dynamin scaffold. The results do not clearly show if endophilin forms short scaffolds in between dynamin assemblies or a continuous scaffold under the dynamin oligomer. Also, there is no attempt to validate the presence of endophilin and dynamin in these structures. Therefore, the colors added to highlight the individual proteins in Figure 3 must be viewed as purely speculative, which should be stated clearly in the text.

In Figure 4 the authors show convincing results indicating that in cells overexpression of endophilin causes significant reduction in the rate of dynamin-dependent transferrin uptake, consistent with the notion that endophilin acts as an inhibitor of dynamin-mediated fission.

In all, this is an interesting and elegant multi-disciplinary study that aims to further our understanding of a fundamental cellular process. The authors provide evidence of the inhibitory role of endophilin in dynamin-mediated fission both in vitro and in situ, and suggest a possible mechanism underlying this function based on structural evidence. However, the majority of the data presented fails to provide further insight beyond what has been previously reported and suggested in studies by Farsad et al. 2001, Sundborger et at 2011, Meinecke et al. 2013 and Boucrot et al. 2013. Therefore, it's the opinion of this reviewer that while the novelty of this study is some-what limited, it serves an important role to confirm previously reported findings and suggested mechanisms.

---

## [Author Response]

Summary:There is a consensus among the reviewers that the in vivo and in vitro experiments clearly show excess endophilin blocks dynamin mediated fission when assembled together on lipid tubes and when overexpressed in cells. This is especially convincing in Figure 2 where they label for both endo and dynamin on the same tube. In these experiments both proteins are added simultaneously. However, in Figure 1 it is not clear if dynamin binds or how much binds to the preassembled endophilin tubes since here endophilin is added prior dynamin. Either fluorescently tagged protein or images of endophilin alone compared to endophilin and dynamin would help convince the readers.

We have added a figure (Figure 1—figure supplement 1) showing that the tubes in bright field in Figure 1 are both coated with endophilin and dynamin.

As suggested by one reviewer, the statement that the Bead experiments shows endophilin blocks dynamin mediated constriction should be rephrased to state that dynamin may constrict without causing bead rotation (supercoiling).

We reformulated the sentence in the following way: “These results showed that endophilin blocked mechano-chemical constriction of dynamin and subsequent membrane fission. However, we cannot exclude that dynamin may constrict through another mechanism.”

The reviewers all agree that the in vivo assays in Figure 4 are convincing and clearly show overexpessing endophilin delays dynamin mediated fission and decreases transferrin uptake in proportion to the levels of endophilin expression.

The reviewers all agree that the in vivo assays in Figure 4 are convincing and clearly show overexpressing endophilin delays dynamin mediated fission and decreases transferrin uptake in proportion to the levels of endophilin expression.

We thank the reviewers for acknowledging the quality of these data.

The EM experiments have caused the most concern. It is unclear if the 2D class averages are from the cryo-EM data or negative stain. The Materials and methods suggest the 2D class averages are from cryo-EM data and if this is the case, stating this in the text and figure legends would validate the statement in the Abstract that cryo-EM was used to show the endophilin intercalates between turns of the dynamin helix. As commented by two of the reviewers, the resolution of the class averages is poor and more evidence is needed to support the model that endophilin prevents dynamin interactions in trans across the rungs of the helix. In addition, more quantification of the EM images would also help support the conclusions such as measurements between the GTPase domains in adjacent rungs, change in helical pitch and average diameter. A GTPase assay would help support the lack of G domain trans interactions.

We thank the reviewers for noting their confusion about which results corresponded with negative stain versus cryoEM imaging. In the revised figures, we have taken care to identify which imaging approach we used in the main text, the legends, as well as the methods. Specifically, in revised Figure 3, the cryoEM results (3A-C) are separated from the negative stain EM results (3D-G). We have also collected additional cryoEM datasets that, despite the significant heterogeneity in the co-complex assemblies, have improved the resolution of 2D cryoEM averages. Please see the details below under Essential revisions point 1.

The reviewers also agree that an expanded discussion of the results, taking into consideration other findings, would be beneficial. For example, how do the results shed light on the role of endophilin in the uptake of Shiga toxin, what is the role of endophilin's amphipathic helix, how does the data relate to physiological conditions and previously been published structural studies.

This is a fair comment from the reviewers, and we have substantially modified the Discussion part of the results (see third and last paragraphs).

In addition, considering the main arguments supporting the conclusions are from cell biology and light microscopy results and not from structural studies, it is recommended that the title be reworded and not include the work 'Structural', which may be misleading and not reflecting the strengths of the paper.

In the light of the new cryoEM results we report in this revised manuscript, we would prefer, if the reviewers and editor approve, to use the title: “Structural inhibition of dynaminmediated membrane fission”. We acknowledge that these are low resolution, 2D structural insights, but the model that we argue best explains our data is a structural model, as schematized in Figure 4: endophilin interleaving and sterically preventing dynamin oligomers from polymerizing into helical collars capable of *trans* interactions between helical turns. In addition to the data reported here, this model and rests on a great deal of 3D structural data reported in the past by many laboratories.

If the use of the word “structural” remains contentious, we propose “Architectural-inhibition” or “Steric-inhibition” instead.

Overall, the reviewers agree that the data is of high quality and value and by incorporating their suggestions, the manuscript will be of interest to a wide audience, especially the fields of endocytosis and membrane remodeling.Essential revisions:1) For the EM data, improve the resolution to support the conclusions and add quantification and measurements.

In Figure 3—figure supplement 1 and Figure 3—figure supplement 2, we report the details of newly obtained cryoEM datasets for dynamin-only and endophilin-dynamin coassemblies. As shown in Figure 3 and Figure 3—figure supplement 1, the molecular details apparent in these updated images and class averages notably strengthen our understanding of how endophilin prevents dynamin-1 from forming helical collars. Specifically, the new cryoEM class averages and quantifications in Figure 3—figure supplement 1 show in molecular resolution how full-length, wild-type dynamin-1 forms 2-start or 1-start helical assemblies stabilized by *trans* interactions between adjacent helical turns. By contrast, endophilin interleaves between helical turns, spacing them apart and preventing *trans* interactions. To complement the raw cryoEM and 2D class averages obtained from cryoEM, we have also included negative stain images in different nucleotide conditions (GMPPCP versus GDP, Figure 3) that, consistent with the prior studies, reveal the same phenomenon: endophilin prevents dynamin oligomers from engaging in *trans* interactions.

2) Expand the Discussion to incorporate other findings and further discuss potential mechanisms of action in vivo.

This is a fair comment from the reviewers, and we have substantially modified the Discussion part of the results (see third and last paragraphs).

Reviewer #1:Manuscript on membrane fission driven by dynamin. It is observed that endophilin inhibits dynamin-mediated tubule constriction and fission on model membranes by reducing fission rate and efficiency. Using cryo-EM, evidence is provided for an intercalation of endophilin between turns of dynamin oligomers. In cells, the overexpression of endophilin inhibits transferrin uptake.The study presents a tour de force approach of modern membrane biophysics, including different model membrane systems, visual inspection of molecular dynamics at the nanoscale, optical tweezer setups, careful quantification, and some cryo-EM and cell biological experimentation. The body of data is quite convincing. The possibly biggest limitation comes from the fact that no evidence could be provided for this mechanism to operate under physiological conditions. The authors seem to be aware of this themselves, as they start the Discussion by stating that "the non-physiological 4x molar excess of endophilin inhibits scission completely by blocking dynamin constriction". They then suggest that "many BAR family proteins with dynamin binding SH3 domains can potentially inhibit membrane fission by co-assembling with dynamin polymers when present at a stoichiometric excess relative to dynamin". Any more direct evidence for this would clearly strengthen the physiological claim of this manuscript in a decisive manner.

This reviewer has identified an important limitation of our study. As discussed in the revised version, we think that because no BAR protein is in strong excess when stoichiometrically compared to dynamin in vivo, this inhibitory effect may not be relevant. However, as stated by this reviewer, many BAR proteins could bind dynamin at the same time, and if (as we stated) all of them have an inhibitory action, the stoichiometry of all BAR proteins should be sufficient in the cell to block dynamin fission.

Our statement that “many BAR family proteins with dynamin binding SH3 domains can potentially inhibit membrane fission by co-assembling with dynamin polymers when present at a stoichiometric excess relative to dynamin” is relying on preliminary data obtained with amphiphysin and the F-BAR protein Fbp17 (not shown). This may be part of another future publication.

Reviewer #2:[…] This is an interesting and scholarly study of high general interest and the data are of excellent quality. I, thus, have only a few relatively minor questions and suggestions for improvement.1) My only point of concern pertains to the (central) proposal that endophilin assemblies intercalate between adjacent dynamin rungs in co-assembled polymers. Given the low resolution of the negative stain TEM I feel that additional evidence for this is required. What is the exact distance between GTPase domains in adjacent rungs? Can GTPase domains assembled on lipid tubes be crosslinked in the presence of endophilin?

Please see the details reported above under Essential revisions point 1, and the cryoEM, not negative stain, images shown in revised Figure 3 and Figure 3—figure supplement 1 and Figure 3—figure supplement 2. The spacing between GTPase domains for dynamin-only structures, 1-start and 2-start, are annotated in Figure 3—figure supplement 1 as nearly identical for the 1-start (13.9 nm) versus 2-start (14.0 nm) assemblies. The cryoEM (Figure 3) as well as negative stain (Figure 3), reveal a variable but variable, but unmistakable doubling of the space between dynamin oligomers by endophilin to 29.4 ± 5.2 nm.

If the model was correct then addition of endophilin should inhibit lipid-stimulated GTPase activity of dynamin in a dose-dependent manner. Has this been tested using the GUV system?

This is indeed the case as previously reported in Farsad et al. JCB 2001. We reproduced these data and have found the same results, which we have included in Figure 1—figure supplement 1.

2) Does inhibition of dynamin-mediated fission depend on the amphipathic helices within endophilin A2?

It depends on the presence of a functional BAR domain (see Figure 2), which indicates that endophilin has to bind to the membrane in order to inhibit dynamin. Since amphipathic helix strongly participate in the membrane binding properties of endophilin (Peter et al. Science 2004), we expect that the inhibitory activity of endophilin would be strongly reduced in the absence of amphipathic helices.

3) How do these data fit with the recent proposal that endophilin facilitates fission, e.g. during CIE of Shiga toxin? This point at least should be discussed. In general I feel many readers would benefit from an extended discussion of the new findings and how they relate to previous studies in the literature.

The reviewer is right, we have thus extended the Discussion part (third and last paragraphs). Importantly, while endophilin1 (or A1) covers membrane tubes uniformly, endophilinA2 forms rigid scaffolds that grow like dynamin helices (see Simunovic et al. PNAS 2016) which mediates fission through a friction based mechanism while the tube is under extension forces (see Simunovic et al. Cell 2017). This effect is thought to be dominant in endocytic pathways independent of clathrin (Boucrot et al. Nature 2015) and Shigamediated endocytosis (Renard et al. Nature 2015). However, in fission of clathrin pits, where both A2 and A1 interact with dynamin, their roles may be different as we discuss in the revised Discussion section.

Reviewer #3:[…] In Figure 1 the authors use an in vitro assay to investigate the effect of endophilin on dynamin-mediated membrane tubulation and GTP-dependent fission. They convincingly show that addition of endophilin results in the formation of tubulates that withstand addition of GTP, suggesting that endophilin inhibits dynamin-mediated fission. They further investigate whether the presence of endophilin also interferes with GTP-hydrolysis-driven constriction of dynamin assayed by rotation of beads in a chamber. The presence of endophilin blocked dynamin-mediated bead rotation, suggesting that endophilin prevents not only fission, but also dynamin-mediated conformational changes. Comments, the authors claim that these results show that endophilin blocks dynamin constriction. However, the assay in question detects coiling of dynamin-decorated lipid tubes, rather than dynamin-mediated constriction. Coiling of dynamin-decorated lipid tubes may or may not have any significance to fission in vivo. Also, recent studies show that constriction of dynamin scaffolds occur upon GTP binding rather than hydrolysis (Sundborger et al., 2014). Therefore, these results should be regarded as evidence of endophilin blocking dynamin-mediated GTP-dependent conformational changes, but not necessarily membrane constriction, which is likely to occur simultaneous to dynamin scaffold assembly.

To answer to the reviewer’s comment, we have reformulated the sentence in the following way: “These results showed that endophilin blocked mechano-chemical constriction of dynamin and subsequent membrane fission. However, we cannot exclude that dynamin may constrict through another mechanism.”

In Figure 2 the authors show inhibition of dynamin-mediated GTP-dependent fragmentation of nanotubes pulled from GUVs in the presence of GTP and endophilin. This inhibition is not observed in the presence of BAR domain only, indicating that the effect is dependent on SH3-PRD interactions between the two proteins. Comment: This assay fails to address the role of endophilin as a membrane remodeling protein. By using already constricted nanotubes, this assay bypasses a crucial step in the dynamin-mediated fission process – membrane constriction. Endophilin, the most likely candidate to promote constriction of the membrane upstream of dynamin recruitment, may regulate dynamin assembly as a function of curvature induction. This concept is even supported by previous work by the authors (Roux et al., 2010). Consequently, when using already-constricted tubes, the authors effectively bias their attempts to define the molecular mechanisms of endophilin-mediated inhibition of dynamin by not allowing it to act as a membrane remodeling protein, but rather as a potential recruiter of dynamin.

We would like to nuance some of the statements made by the reviewer above. It is true that in the assay used in Figure 2, we do not directly address the role of membrane remodeling action of endophilin in the process. However, we have fully characterized with this tube pulling assay the remodeling properties of amphiphysin (Sorre et al. PNAS 2012), which is close enough to endophilin to say that at the concentrations we used in this assay, endophilin most probably participate in the tube constriction, and acts as a membrane remodeling protein. It may thus participate in dynamin recruitment to an already pre-constricted tube as proposed by the reviewer, however in this configuration, dynamin cannot fission the tube.

As a matter of fact, in Figure 1, we show time-lapses images of endophilin-dynamin coated tubules onto membrane sheets that were initially formed by addition of endophilin, and then subsequent addition of dynamin. In this case, dynamin is specifically recruited to the tubules formed by the membrane remodeling action of endophilin. This may have a physiological role (see Discussion, fourth paragraph).

In Figure 3 the authors use cryo-EM to visualize the endophilin-dynamin complex assembled on lipid tubes. These results show the impact of endophilin on dynamin scaffold assembly. Comments: The 2D class averages presented in this figure in the key point of the paper and should be emphasized. This data is confirming previously suggested mechanisms of endophilin inhibiting fission by preventing proper dynamin assembly. The resolution of the images does not support such claims, which should be clearly stated in the text. Furthermore, there is no evidence that endophilin actually interleaves the dynamin scaffold. The results do not clearly show if endophilin forms short scaffolds in between dynamin assemblies or a continuous scaffold under the dynamin oligomer. Also, there is no attempt to validate the presence of endophilin and dynamin in these structures. Therefore, the colors added to highlight the individual proteins in Figure 3 must be viewed as purely speculative, which should be stated clearly in the text.

Please see the details reported above under Essential revisions point 1, and the cryoEM, not negative stain, images shown in revised Figure 3 and Figure 3—figure supplements 1 and 2.